# The General Pedagogical Knowledge Underpinning Early Childhood Education Teachers’ Classroom Behaviors Who Teach English as a Foreign Language in Chinese Kindergartens

**DOI:** 10.3390/bs14070526

**Published:** 2024-06-24

**Authors:** Xiaobo Shi, Susanna Siu-Sze Yeung

**Affiliations:** 1Department of Early Childhood Education, Henan Normal University, Xinxiang 453007, China; 111099@htu.edu.cn; 2Department of Psychology, The Education University of Hong Kong, Hong Kong SAR, China

**Keywords:** early childhood English education, GPK, English as a foreign language, novice teachers, experienced teachers

## Abstract

There is limited understanding of the general pedagogical knowledge (GPK) of early childhood education (ECE) teachers who teach English as a foreign language (EFL). This study therefore explored GPK categories and subcategories in six Chinese ECE EFL teachers using stimulated recall classroom observation. The deductive and inductive data analysis revealed that GPK consists of four categories and ten subcategories. The four knowledge categories were child development, the objectives and content of teaching, the act of teaching, and classroom management. Additionally, the study found that the novice teachers had similarities with the experienced teachers in number and type of GPK. The study also found differences: the subcategory how to use child-appropriate instructional methods was top for the experienced teachers, but not for the novice teachers; the novice teachers mentioned classroom management more than the experienced teachers; the novice teachers showed a negative tendency toward the act of teaching, while the experienced teachers were positive. The implications are discussed.

## 1. Introduction

In the area of teacher research, the paradigm has shifted from emphasizing teachers’ behaviors to teachers’ cognitions or knowledge in recent decades. Drawing inspiration from Shulman’s seminal work, researchers have strived to categorize teacher knowledge into various domains, including pedagogical content knowledge (PCK), content knowledge (CK) and general pedagogical knowledge (GPK), across different subject areas and educational levels [1]. Pioneers in the field define these dimensions as follows: CK encompasses fundamental concepts and principles within a subject that guide the framing and evaluation of statements in that area; GPK encompasses teaching skills and includes knowledge that is distinct from content expertise; PCK encompasses a teacher’s understanding of adapting instructional material to meet the needs of diverse classrooms and educational settings [1,2]. In most cases, these investigations center on CK and PCK. GPK has been recognized as an overlooked area of teacher knowledge [3]. As defined by Shulman, GPK is involved with knowledge of the principles and techniques of classroom management, learners and learning, assessment, and educational contexts and purposes [1]. A thorough literature review has revealed that contemporary research on teachers’ GPK is related to three broader areas: knowledge of student learning, such as students’ individual personalities and learning processes; instructional processes, such as classroom management and teaching methods; and assessment, such as evaluation procedures and diagnostic approaches [4,5].

In the field of teaching English as a foreign language (EFL) or a second language (ESL) education, various contexts have been explored in the research on GPK, such as writing, reading, and speaking instruction. However, all of these studies have taken place in adult language education settings. For example, Tsui conducted research on four ESL instructors focusing on writing instruction in Hong Kong [6]. The study found that the novice teachers perceived classroom management as a strategy for maintaining order among students, while experienced teachers believed it aided in the achievement of instructional objectives. Abdelhafez investigated three experienced EFL teachers in Egypt secondary and preparatory schools and identified two core areas of GPK: approaches to classroom management and content and task management. Content and task management includes aspects such as lesson planning, time management, and giving instructions [7]. In Gatbonton’s study on the knowledge base of ESL teachers, the novice teachers primarily focused on observing student behavior and reactions rather than language management, with less emphasis on detailed plans and broad objectives in comparison to experienced teachers [8]. In a study by Karimi and Norouzi, the growth of four novice ESL teachers’ pedagogical knowledge base was examined through mentoring initiatives led by four experts. Findings indicted that the novice teachers commonly focused on student behavior before the intervention, while in their post-mentoring instruction, language management and knowledge of students were prominent [9]. Tajeddin and Bolouri conducted a comprehensive investigation on the pedagogical knowledge of two inexperienced EFL teachers working at private language institutes. Findings indicated that the inexperienced teachers utilized their pedagogical knowledge to reach curriculum goals and manage the students’ misbehaviors, but not students’ learning [10]. These studies, using qualitative methods such as stimulated recall classroom observations, have explored differences between novices and experts in terms of general knowledge. They offer valuable perspectives, highlighting that classroom management, lesson planning, and knowledge of students are integral aspects of GPK for EFL or ESL teachers.

In recent years, an increasing number of researchers have employed quantitative methods, such as tests, to study the structure of EFL or ESL teachers’ GPK. A team in Germany, led by Johannes König, developed a test to measure EFL teachers’ GPK. This test assesses knowledge in four areas: preparing, structuring, and evaluating lessons (“structure”); motivating and supporting students, and managing the classroom (“motivation/classroom management”); handling diverse learning groups (“adaptivity”); and assessing students (“assessment”) [3,4,11,12,13]. These studies indicate that student assessment is an important component of EFL teachers’ GPK. Despite the qualitative and quantitative studies, there is a lack of definitive findings from empirical studies regarding the characteristics and structure of GPK for EFL teachers in the field of early childhood education (ECE). To address this gap, this research examined whether these components apply to Chinese ECE teachers in terms of their GPK.

In ECE, a number of GPK components in EFL or ESL education, such as classroom management, lesson planning, and knowledge of child development, have been buttressed by several studies of foreign language teaching for young learners [14,15,16]. As the language acquisition process and cognitive development of young children are qualitatively different from those of adults, ECE EFL teachers may have some specific components in GPK that are unique to this professional filed. They are more likely to value the classroom environment, or the level of motivation children exhibit in their learning experiences. It is also important for ECE educators to have a detailed plan for each lesson and to use developmentally appropriate practices. For example, Nunan highlighted the importance of creating conductive learning environments for classroom management at lower grade levels [17]. Nikolov found the youngest children were intrinsically motivated by teachers and classroom environments. Building a rapport with children facilitates a positive classroom environment, thereby encouraging them to learn English [18]. Given that early childhood language teachers have limited class time, effective time management and well-structured lesson plans are crucial. The early language teachers need to have another important ability, that is how to consider learners’ levels. Butler examined elementary teachers’ struggles with communicative activities, finding they had difficulties in designing developmentally appropriate activities for children. It was concluded that utilizing activities that were not aligned with the children’s developmental levels proved to be inefficient [19].

Through examining relevant research, GPK in ECE EFL teachers is conceptualized as having three components: classroom management, lesson planning, and knowledge of child development. Simultaneously, ECE EFL teachers’ GPK may exhibit unique characteristics within this specific educational setting. However, the empirical studies examining the GPK of ECE EFL teachers are limited. A notable exception is Kim’s study, which surveyed 336 ECE EFL teachers in Korea using a teacher knowledge questionnaire and confirmed this three-component GPK model [20]. Overall, there is a need to systematically conceptualize, measure, and validate these components [4,11,20]. Also, many scholars have highlighted the ongoing debate over differentiating GPK from PCK. They argue that GPK, while at a different level, has its own distinct content and should not be confused with or reduced to PCK. This emphasizes the need for further empirical studies to clarify this distinction [4,5]. This study therefore aims to explore the presence of GPK underpinning the classroom behaviors of ECE EFL teachers in China to enhance our understanding of its conceptual boundaries and practical implications. This will contribute to a more comprehensive understanding of GPK in EFL/ESL contexts across different education systems.

While there is a wealth of comparative research on novice and experienced teachers within the broader realm of teacher education, such investigations remain limited within the specific domain of EFL or ESL education [10,21]. Following Gatbonton’s research, a few scholars have conducted in-depth comparative analyses of the dominant GPK categories and their prioritization between novice and experienced EFL or ESL teachers. This line of research offers a fresh perspective for comparing the two groups. In her pioneering study, Gatbonton set a 6% cut-off point for a knowledge category to be included in the list of dominant categories. Using this criterion, her research found some similar dominant pedagogical knowledge categories among novice and experienced ESL teachers, such as language management, knowledge of students, and noting student behavior and reactions. However, the number and rank order of dominant categories differed between the two groups. Novice teachers had nine dominant categories, whereas experienced teachers had six. For experienced teachers, language management topped the list, while for novice teachers, noting student behavior and reactions was the most prominent [8]. To extend this line of research, Karimi and his team conducted a series of comparative studies on the dominant pedagogical knowledge categories of novice and experienced ESL teachers [9,21]. For example, one of their latest studies found that both novice and experienced ESL teachers focusing on reading instruction reported similar dominant categories, such as language management and aiding reading comprehension. However, the frequency and ranking of these dominant knowledge categories differed. The findings revealed that experienced teachers produced pedagogical thoughts twice as often as novice teachers and had significantly more thoughts about language management and aiding reading comprehension [21].

It is rare to find research investigating both sets of ECE EFL teachers together in one study and comparing them on very specific points. Therefore, this study also tried to identify more clearly how these two sets of teachers differ or how they are similar to each other in GPK, following this line of research.

### 1.1. Teaching English to Young Children in China

In the educational landscape of China, an expanding number of kindergartens now offer early English language learning as an integral component of their curriculum. Nevertheless, these kindergartens exhibit significant diversity with regards to their educational programs and instructional approaches. These kindergartens normally design their own curricula. The amount of time allotted for English instruction also shows variability. In some kindergartens, the English instruction time is less than two hours each week. In some kindergartens, Chinese and English each take up 50% of the instruction time in order to establish a bilingual learning environment [22].

In Yu and Ruan’s research, they identified two prevalent categories of English teachers in Chinese kindergarten classrooms [23]. The homeroom teachers, who typically hold degrees in ECE, exhibit systematic theories and skills within the realm of early childhood education, curriculum and psychology, but lack sufficient proficiency in English. The teachers who are the specialists tasked with teaching English to kindergarteners, who typically hold degrees in English, excel in English proficiency but lack sufficient expertise in early childhood education. The teaching practices and techniques they implement may not effectively cater to the growth and learning requirements of kindergarten students. Additionally, in a study conducted by Tang, it was revealed that almost half of the teachers in Xi’an, China, had less than three years of teaching experience [24]. In Tianjin, China, according to Gao’s research, almost 80% of teachers had less than five years of experience in teaching [25].

Since 2018, the Ministry of Education (MOE) has imposed a strict prohibition on the provision of English language classes in kindergartens. There is a strong demand for more empirical studies which can inform policy makers to regulate the field properly in China [22,26]. In order to achieve success in early childhood English education in China, it is crucial to properly train teachers who possess strong professional abilities and a solid knowledge base [27]. For this purpose, the current research aimed to explore the GPK of ECE EFL teachers in China.

### 1.2. Research Questions

This study used a qualitative research approach to examine the structure of ECE EFL teachers’ GPK in Chinese kindergartens, and to compare GPK of experienced and novice teachers. This study aimed to answer two research questions:What are the categories and subcategories of GPK of ECE EFL teachers underpinning their classroom behaviors in China?What are the similarities and differences in GPK categories and subcategories between the experienced and the novice teachers?

## 2. Methods

### 2.1. Research Sites and Participants

To choose the target kindergartens, we followed these criteria: (1) the kindergartens have been accredited as first-level institutions by local educational administrative departments; (2) the kindergartens have been running English teaching for several years in order to establish a well-developed and consistent English curriculum structure and English teaching staff; (3) the kindergartens are located in capital or prefecture level cities. Based on these criteria, we selected four kindergartens from a pool of potential candidates. Kindergartens A, B and C are located in Zhengzhou, Henan province (This province, located in Central China, with a medium level in early childhood English education, is a relatively representative province of China. According to the Henan Education Statistical Bulletin, approximately 750 kindergartens offer English curricula). A, B, and C are considered the top private kindergartens for English teaching in the province, with identical English curriculum frameworks designed by the same company (Since 1996, this company has been operating more than 100 kindergartens identified as conducting early English language education, which are located in many provinces of China. These kindergartens are well-known and relatively representative of early childhood English education in China). We selected Kindergarten D, a public kindergarten affiliated with a foreign language university and located in Luoyang, Henan province, to increase the diversity of the sample. This kindergarten has been operating its school-based English program since 2000 and been regarded as the best public kindergarten for English teaching in the province.

In selecting our sample, we intentionally chose six individuals with varying backgrounds from the four kindergartens. We considered their experiences teaching English to young children as a key factor. Candidates with over five years of experience are considered as the experienced teachers, while those with less than three years are novice teachers [8]. The principals’ recommendations and the English teachers’ willingness were also taken into consideration. After obtaining permits from the Education Department of Henan Province, all principals, participants and the children’s parents signed consent forms prior to the classroom observations. Ultimately, six female teachers took part in the study. Table 1 details the characteristics of the six sampled teachers, consisting of two novice teachers and four experienced teachers.

### 2.2. Data Collection

We employed stimulated recall classroom observation to collect data about ECE EFL teachers’ behaviors and their underpinning knowledge. Previous investigations have utilized this kind of observation to study teacher knowledge [7,8,9,10]. In one paper published by the two authors, the PCK of ECE EFL teachers was investigated using this specific method [28].

#### 2.2.1. Videotaped Classroom Observation

The first author carried out open-ended classroom observation to investigate ECE English classroom teaching in Chinese kindergartens. Researchers have identified some ECE EFL teachers’ developmentally appropriate oriented behaviors, such as providing a variety of (1) English teaching materials and English picture books, (2) teaching aids (e.g., flash cards, realia), (3) instructional activities or methods (e.g., games, singing), and (4) rewarding and evaluating children’s learning effects, (5) planning and conducting lessons (e.g., integration of kindergarten curriculum with English lessons, teaching processes), and (6) the usage of English [16,29,30]. Thus, the observation focused on these categories to construct ECE EFL teacher knowledge underpinning their practice in the Chinese context.

When studying teacher knowledge, classroom observation can provide a benchmark through which the relationship between what teachers know and how they apply the knowledge in classroom practice can be reflected [7,9]. To minimize the effects on teachers and children, the first author scheduled a familiarization period with preliminary visits before formal videotaping. Additionally, the first author sat in a remote corner of the classroom to avoid affecting the teacher’s instruction and the children’s learning throughout the study. It took about two months to finish the observation. The first author observed twenty-four classes led by the six teachers (K1, a class with 3–4-year-olds, eight lessons; K2, a class with 4–5-year-olds, eight lessons; K3, a class with 5–6-year-olds class, eight lessons).

The frequency of observations was determined based on various factors, including the frequency of English lessons scheduled by the kindergartens (usually twice a week) and the teachers’ daily work. Within a span of two weeks, we taped a total of twelve lessons taught by Teacher H and Teacher E, with each teacher conducting six lessons, including two K1 lessons, two K2 lessons, and two K3 lessons. We taped a total of twelve lessons for the other four teachers over a three-week period. Each teacher held three lessons, including one K1, K2 and K3 lesson respectively.

There are numerous benefits to collecting English lesson material over a span of two to three weeks. We could outline how teachers carried out English teaching activities with different themes and how they interacted with children across different ages and with varying levels of development. Furthermore, we could assess the continuity of English teaching content by observing lessons across consecutive weeks. The first author rewatched all the lessons on video and captured additional insights through detailed field notes to enhance the video documentation of the lessons.

#### 2.2.2. Stimulated Recall

After observing each lesson, the first author engaged in a stimulated recall session with the teacher on the same day to discuss the lesson in detail. She asked the teachers to watch and review all the recorded lesson videos, paused them at certain points, and let the teachers articulate their rationale underpinning certain behaviors. The teachers were able to actively perceive the intricate nature of their cognitive processes or knowledge during classroom practice by using this kind of stimulated recall protocol. Meanwhile, this could aid the teachers to describe and analyze their teaching process specifically and completely [9]. As such, we were able to pinpoint the specific knowledge categories or subcategories talked about and practiced by the teachers [7,8,9,10].

To enhance memory retrieval, the first author minimized the time gap between teacher thinking while teaching and the reporting after the teaching, and always conducted the stimulated recall immediately after the observation. Gass and Mackey recommended this approach to ensure its reliability [31]. We captured audio recordings of all the stimulated recall interviews that took place after classroom observations. These made up the foundational materials for delving into the teacher knowledge that informs their classroom behaviors.

The first author carried out a pilot study to trial the interview schedule with one English teacher (not part of the primary sample) from kindergarten C. She constructed a preliminary interview schedule. The questions used in the stimulated recall were concerned with six areas of teachers’ classroom behaviors, including teaching materials, teaching aids, teaching activities, evaluating and rewarding techniques, integration of curriculum, and English usage (see Table 2). Specific questions were modified based on actual classroom observations.

Following each stimulated recall session, the first author transcribed the audio recordings within the same week as the observation took place. In addition, she performed initial evaluation of the transcription in order to identify any issues necessitating further investigation. These became the questions of subsequent interview sessions. She conducted this kind of follow-up interview with each participant until the teacher reached a point of inability to offer any additional information. In other words, the data had reached a point of saturation [32,33].

### 2.3. Data Analysis

We used NVIVO 12 Pro to analyze the data. Through reviewing the audio transcription of the stimulated recall sessions, a summary of each session was compiled. Altogether, the duration of the observations was approximately ten hours. The interview stretched over a span of approximately 14 h and 273,392 words were collected. The two authors collaboratively analyzed the data. Inter-rater agreement on the coding was sufficient to ensure the validity of the codes (ICC = 0.85). Based on the literature review, GPK in ECE EFL teachers is conceptualized as consisting of three components: knowledge of child development, lesson planning, and classroom management [4,20]. Therefore, we utilized the subsequent framework to analyze the qualitative data: (1) Knowledge of child development underpinning classroom behaviors (episodes which revealed this category of knowledge and its impact on the teacher’s pedagogical choices), (2) Knowledge of the objectives and contents of teaching underpinning classroom behaviors (episodes which revealed this category of knowledge and its impact on the teacher’s pedagogical choices), and (3) Knowledge of classroom management underpinning classroom behaviors (episodes which revealed this category of knowledge and its impact on the teacher’s pedagogical choices).

After coding all the transcripts of the stimulated recall data one by one, we divided the data into meaningful segments and synthesized them with categories and subcategories. Upon giving codes to the categories and subcategories of the first stimulated recall session analyzed, we applied these codes to the remaining stimulated recall sessions. Additional codes were developed as the analysis progressed. In repeated examination of the stimulated recall data, we identified a category that was not included in the above framework, i.e., knowledge of the act of teaching underpinning classroom behaviors (episodes which revealed this category of knowledge and its impact on the teacher’s pedagogical choices). We included this theme and analyzed all the data inductively (see Table 3). After this, we compared the categories and subcategories of the novice teachers to those of the experienced teachers (see Table 4). We first compared the teachers in terms of the number of GPK categories and subcategories and how frequently these occurred. Later, we focused upon the top dominant GPK categories and subcategories among both groups of teachers.

## 3. Results

### 3.1. ECE EFL Teachers’ GPK Categories and Subcategories Underpinning Classroom Behaviors

The findings of the study identified four key areas of GPK essential for ECE EFL teachers in China: knowledge of child development (40%), knowledge of the objectives and content of teaching (24%), knowledge of the act of teaching (24%), and knowledge of classroom management (12%). The percentage of each category was calculated by dividing the number of references in that category by the total 728 references (see Table 3 and Figure 1 for details). The core categories and subcategories are discussed in the following section.

**Child development (40%)**. Early childhood language teachers’ various components of knowledge of children function together to influence their classroom behaviors, such as knowledge about child learning processes, familiarity with individual children and the learning objectives tailored to children’s needs [34]. This category encompasses how to use child-appropriate activities, how to address individual developmental differences, and how to use child-appropriate instructional methods.

First, the participants’ views regarding how to use child-appropriate activities focused on three types of activity organization: teacher–whole class, corner activities (activities in an interest corner) and teacher–individual child. The current English teaching in Chinese kindergartens predominantly consists of teacher–whole class activities. For example, one participant pointed out the reason why she chose the teacher–whole class activities in a review lesson. All the children were required to master the words in a review lesson. Some children might have mastered this, and some might still find the words to be strange. She “*should first finish teaching the words to the whole class*” and “*may question the children one by one to check if each child has grasped the words at the end of the class*” (Teacher H). There were few corner activities related to English learning; these were sometimes carried out after class. As pointed out by one participant: “*I taught color in this lesson and would put some colored water and plasticine in the art corner. Some children may only use colored water in class, and they can use the plasticine in the corner after class*” (Teacher E). Similarly, individualized instruction usually took place after class: “*After class, I will focus on guiding this child. However, in class, I could not always give my attention to this child. The other children were still waiting*”. (Teacher R).

Second, the participants documented children’s developmental differences in language, cognition, motor, emotion and sociality. The teacher–child interactions were affected by children’s individual differences in language development: “*When I asked, ‘What is this?’ some children could directly say ‘puppy,’ some needed to be reminded of the word ‘puppy’, and some had no response even after my reminding them again and again*” (Teacher H). Likewise, the influence of children’s cognitive and motor development levels on English learning and teaching was evident: “*In lesson number two, the K1 children may not have reached the level of establishing a connection between numbers and objects. If they know what the numbers mean, they may understand it in English more easily*” (Teacher R). As regards emotional and social development, children’s differences in personality characteristics in different classes were mentioned by some participants: “*The more people are there, the more introverted they [the children in another class] are. They are afraid to speak English. They are not like the children in my class, who are outgoing*” (Teacher R).

Children’s differences in English learning experiences were also pointed out by the participants: “*Like these two babies, who were new students. I had to say the instruction many times or give many examples so that they could do the same action like me*” (Teacher E). The participants finally confirmed children’s differences in age characteristics: “*Children have age characteristics in each grade. In K1, a child may rely more on observation to do something. ‘I [the child] see it and then I [the child] do it’. In K2, ‘I do what I see and what I hear’. When in K3, ‘I do what I hear’. A child has such a gradually developing process*” (Teacher E).

Third, the teachers based their child-appropriate instructional methods on children’s developmental differences. Nine child-appropriate instructional methods were proposed by the participants: scaffolding, play-based, aptitude-based, child-centered, teaching in an orderly way, peer learning, teaching in situation, following the law of children’s development, and simplifying what is complicated. For instance, one participant described how she used different questioning strategies, embodying the concept of teaching students according to their aptitudes: “*Some children learn faster and raise their hands positively. I do not name them. I say ‘Ok, close your eyes, close your mouth’. I name those who are not raising their hands or who are not positive*” (Teacher W). Another participant explained her understanding of teaching in situation: “*The bear was tired after walking. He needed to sit down and have a rest. This led to the sentence ‘Sitting in the chair’. Then, he was so tired and thirsty, he needed to eat some fruit, so he ate a pear, and this led to the sentence ‘Eating a pear’*” (Teacher E).

**The objectives and contents of teaching (24%).** This category included two issues: selection of teaching content and design of teaching objectives. Selecting and creating teaching content and goals are fundamental aspects of lesson planning, which allow teachers to incorporate a variety of further decisions into the classroom [12]. All participants indicated that they selected the educational materials presented in class from the kindergarten textbooks. The four participants from kindergartens A, B and C stated that the main teaching content was “Star English”: “*I follow the lesson plans of Star English designed by our company. If I want to add something new, I go online and search for some simple songs or chants*”. (Teacher S). Likewise, the participants from kindergarten D described the teaching materials of their kindergarten, “Kindergarten Happy English”. “*We developed a set of teaching materials for K1, K2 and K3. They include children’s and teachers’ books. The lesson big bear was selected from the textbook*” (Teacher E).

The participants also emphasized the importance of how to design teaching objectives. The horizontal relationships among different teaching objectives were illustrated by the participants: “*Children can perceive and sing the song Autumn, which is a cognitive objective; children can say the words with accurate pronunciation, which is an ability objective; children can feel the joy of the games, which is an emotional objective*” (Teacher W). Three principles for designing the objectives were also constructed by the participants: to design from the perspective of children, to cover the whole teaching process, and to integrate different curriculum areas. For example, one participant elaborated the principle of how to cover the whole teaching process with objectives: “*The three objectives are actually blending together and running through the entire teaching process*” (Teacher E).

**The act of teaching (24%).** This category includes two main components: the achievement of the teaching objectives and the adjustment of the teaching process. Studies on teacher expertise have found that the expert teachers can closely synchronize their planning choices with the requirements of their students or can carefully assess the suitability of teaching methods to be integrated into the classroom environment of a specific group of learners, in contrast to their less experienced counterparts [12]. For the teaching objectives, it was indicated whether or not the teaching process revolved around the objectives. For example, one novice teacher stated: “*I designed this lesson with two goals: to master the word ‘dog’ and to master the sentence ‘This is a dog’. The first one was finished in class. However, the second one was forgotten and not finished in class*” (Teacher S).

The need was highlighted to adjust the teaching process according to children’s reactions in class, such as their interactions with the teacher, their facial expressions, and the degree of their involvement. For instance, one participant stated: “*I use the flash cards to test whether the children have mastered the four words taught before [red, yellow, blue and green]. If many children are not proficient in these words [no or little response], I will change into the new-content-taught lesson, and abandon the old-content-reviewed lesson [two types of lessons designed by the kindergarten]. The games and materials used in class will be totally different*” (Teacher E).

**Classroom management (12%).** It is crucial for teachers to possess the necessary skills and expertise in effectively managing their classrooms, as it directly contributes to their professional competence [12,14,17]. The participants’ main concerns in relation to classroom management were with three issues: how to manage student behavior, how to gain and maintain student attention, and how to promote motivation for language. Three approaches to manage student behavior were constructed: establishing rules and routines, making use of instructions, and using chants. For instance, one participant revealed different kinds of instructions adopted in class to manage children’s behaviors: “*If I want the children to sit down and listen carefully, I will use instructions, such as ‘One, two, three, four, four, four!’ or ‘Look at me, Chua, Chua, Chua’*” (Teacher S).

Five methods to gain and maintain student attention were highlighted: stickers, games, eye contact, integration of scientific elements and design of instructional activity. According to one participant, the integration of scientific elements could stimulate children’s curiosity so that their attention was maintained. She explained: “*This is a K3 class. I can put in some scientific elements to stimulate their curiosity. For example, let them know where ‘orange’ comes from. There need to be a red and a yellow. Oh, this is amazing! Red and yellow can make orange*” (Teacher H-K3-Orange).

As the third issue, the participants highlighted eight ways to foster children’s motivation to learn English: finding children’s interests, games, stickers and cards, peers, verbal or gesture encouragement, competition games, English Awards Ceremonies, and songs. For example, one participant justified the aims of using verbal or body language to motivate the children: “*Except “Give me five”, there are “Give me hug”, “Give me kiss”, “Give me one finger”, “Give me two fingers”, and “Give me three fingers”. All these are OK. The purpose is the same. The child does not mind what kind of encouragement the teacher gives him/her. What the child minds is whether the teacher notices him/her, whether the teacher gives him/her another encouragement, and whether the teacher sees his/her progress*” (Teacher E).

### 3.2. The Similarities and Differences between Novice and Experienced Teachers

#### 3.2.1. The Similarities between the Two Groups

The data presented in Table 3 indicate that both groups of teachers demonstrated similarities regarding the number and type of GPK categories (4) and subcategories (9–10). Notably, an additional GPK subcategory was observed in the data of novice teachers, specifically relating to how to gain and maintain student attention. Following this, a Spearman rank correlation test was performed on the frequency counts of the GPK subcategories identified in both groups of teachers. The analysis revealed a significant correlation between the two groups (*r_s_* = 0.82, *p* = 0.004), suggesting similarity on this matter. In addition, the most frequently reported GPK subcategory was how to address individual developmental differences for both groups of teachers. However, important differences did emerge between them; these are described next.

#### 3.2.2. The Differences between the Two Groups

A Spearman rank correlation test was conducted on the frequency counts of the GPK categories between the two groups of teachers. The results showed that the correlation is not significant (*r_s_* = 0.80, *p* = 0.200), suggesting differences on this matter. Despite the similarities in the quantity and types of GPK categories within both groups mentioned earlier, there is a distinct difference in the overall pattern. The experienced teachers prioritize the knowledge of child development, whereas the novice teachers pay more attention to the knowledge of classroom management.

In Table 4, the ‘All’ column displays the frequency with which the GPK subcategories were mentioned by each group of teachers, with superscripts indicating dominant subcategories with a frequency of at least 6% [8]. Six dominant subcategories for the novice teachers were yielded, compared to seven for the experienced teachers. It was discovered that there was one additional dominant GPK subcategory present in the experienced teachers’ data compared to the novice teachers, namely, how to use child-appropriate instructional methods.

Upon thorough analysis, it is clear that the dominant GPK subcategories of the two teacher groups yielded different rankings. The top result for novice teachers was how to address individual developmental differences (30%), followed by the adjustment of the teaching process (21%), selection of teaching content (17%), how to promote motivation for language (11%), the achievement of the teaching objectives (7%), and design of teaching objectives (6%). The top result for experienced teachers was how to address individual developmental differences (29%), followed by selection of teaching content, the adjustment of the teaching process (each at 14%), how to use child-appropriate instructional methods, design of teaching objectives (each at 11%), the achievement of the teaching objectives, and how to promote motivation for language (each at 7%) (see Table 4 and Figure 2).The Spearman rank correlation analysis showed that there was no significant correlation between the dominant subcategories within the GPK dataset (*r_s_* = 0.79, *p* = 0.059). This finding indicates that while the dominant GPK subcategories were consistent in both groups in terms of number and type, their order of frequency varied significantly.

## 4. Discussion

### 4.1. The Structure of ECE EFL Teachers’ GPK

The study participants identified GPK as essential knowledge for ECE EFL teachers, which aligns with previous research findings [7,13,35]. According to Abdehafez, utilizing GPK allows teachers to enhance learner engagement to its fullest potential, resulting in a greater capacity for student-centered education [7]. The four categories identified in the study align with Shulman’s initial definition of GPK and the expanded framework of EFL teachers’ GPK outlined and verified by König and his team, which consists of knowledge of student learning, instructional processes, and assessment [1,3,4,11,12,13]. It can be argued that child development is involved in ECE EFL teachers’ mastery of kindergarteners’ learning. The other three categories pertain to their knowledge of instructional processes. This study also validated Kim’s three-component model of ECE EFL teachers’ GPK, with the addition of one extra component: the act of teaching [20].

Mastery of knowledge of students is a fundamental aspect of teacher expertise that is extensively discussed in academic research [36,37]. The data revealed that ECE EFL teachers transformed knowledge of students into classroom behaviors by adapting the objectives, content, processes of teaching, and instructional methods to attend to the children’s needs. This finding is consistent with the research of Hill and Chin, who maintained that a teacher’s knowledge of students is essential for implementing a variety of effective classroom strategies, such as adjusting the pacing of instruction based on student need, assessing student understanding and misunderstanding in the moment, and designing tasks and questions to further student understanding [38]. The newly discovered “dynamic” component, the act of teaching, which encompasses deviating from the lesson plans and adapting the teaching methods, accordingly, has been examined in EFL and ESL education literature. The teacher knowledge research has demonstrated that these deviations are a direct consequence of the ongoing interplay between teachers’ teaching methods and their interpretation of the learning setting, particularly of the students, at any particular time [10,13]. Finally, a qualified ECE EFL teacher should possess a combination of instructional and managerial knowledge. This knowledge not only can scaffold an ECE EFL teacher’s instruction, but also can be a benchmark to distinguish a novice or an expert teacher [12,14].

Another interesting finding is the lack of evaluative knowledge among these ECE EFL teachers. This contrasts with the quantitative studies by König and his team, which found assessment to be an important dimension of EFL teachers’ GPK [3,4,11,12,13]. It may be that teachers of young English learners generally have limited training or experience in assessing this particular age group in many contexts [39,40]. Nonetheless, the extensive literature suggests assessment to be a component of teacher knowledge, which could be the driving force for teachers to maximize student performance, or maintain or increase student interest [4,41]. Another reason might be that this study adopted a qualitative research method and had a small sample size. Future research could use quantitative methods, such as questionnaires or tests, to investigate the structure of ECE EFL teachers’ GPK with a bigger sample.

### 4.2. Comparisons of GPK between Novice and Experienced Teachers

#### 4.2.1. Similarities between the Two Groups

Despite their limited teaching experience, the study revealed that the novice teachers shared similar numbers and types of categories and subcategories of GPK with the experienced teachers. This evidence is presented in the following four areas: four GPK categories; ten GPK subcategories; six dominant GPK subcategories (experienced, seven), and the most frequently reported subcategory (how to address individual developmental differences). The resemblance between the novice and experienced teachers in these aspects implies that even early in their careers, novice teachers have already accumulated a significant amount of the knowledge typically possessed by experienced teachers. This discovery is corroborated by other studies [8,9,10].

This could potentially be linked to the origins of their knowledge, such as in-service training, and learning from peers. The novice teacher H stated: “*Before I came here, I had no contact with children and did not know how to interact with them. After the training [games, pronunciation, teaching methods], I can successfully complete an English lesson*”. The novice teachers also revealed the value of learning from their peers in updating their knowledge, “*Tina has been teaching for eight years. She knows how to interact with children, how to give them thumbs-up. I observe all of these things and learn a lot from her unconsciously*” (Teacher S).

How to address individual developmental differences was found to be a common concern among both the novice and the experienced teachers in this study. This differs from previous studies on EFL or ESL teachers’ dominant knowledge categories. For example, Karimi and his team found that language management was the most frequently reported category. In their study, language management includes engaging students in identifying phrase and sentence structures, understanding word meanings, collectively eliciting and comparing outputs, and providing adaptive corrective feedback [21]. However, this is in accord with previous research of early childhood language education scholars, and may be related to the characteristics of the ECE field [34,36,42]. Schachter found that early childhood language teachers’ classroom practices are, to a great extent, influenced by their knowledge about how children grow, develop and learn [34,42]. According to Canh, knowledge of the learning processes of the young children and how they acquire English is more crucial for effective English language teaching than theoretical knowledge about language and teaching methods [36].

#### 4.2.2. Differences between the Two Groups

The research uncovered numerous differences between the two groups in dominant GPK subcategories and their frequencies. These findings align with previous studies by Gatbonton, Karimi and their teams, extending this line of research into the ECE EFL context [8,9,21].

One notable teacher group difference is that the subcategory how to use child-appropriate instructional methods was top for experienced teachers and for three of the four experienced teachers, but not for novice teachers. A finer-grained analysis of this subcategory revealed that all nine child-appropriate instructional methods were proposed by the experienced teachers. Moreover, compared with novice teachers, this subcategory of experienced teachers is integrated with one other subcategory, how to address individual developmental differences. The experienced teachers based their child-appropriate instructional methods on children’s developmental differences. The experienced teachers used knowledge of learners in a dynamic manner. They paid more attention to how to use the knowledge of learners rather than what the learners know in classroom practice [7]. This can also be observed from the positive interactions between them and children and the high consistency between their GPK and classroom practice.

Another interesting difference is the category knowledge of classroom management. Novice teachers discussed this more in all subcategories than that of experienced teachers. This emphasis on student behavior and attention aligns with research findings in EFL/ESL education [8,9,10]. From observations, it was also found that novice teachers spent a lot of time in English classes on managing children’s behavior so as to maintain the order of the class. For example, Teacher S repeatedly used instructions, “One, two, three, four, four, four!” or “Look at me, Chua, Chua, Chua” to tell children to sit down and listen carefully. However, the experienced teachers focused more on teaching content and pedagogies. For example, Teacher W stated: “*In order to effectively use time to the teaching of key points, I definitely don’t want to spend too much time in building routine in English classes*”. This inclination for the novice teachers to prioritize classroom management can be explained by the fact that experience is an important source of teacher knowledge [7,10,12]. As reflected by Teacher W: “*The ‘accumulation’ of my work experience-I think it refers to two aspects: one is an unspoken agreement between children and me; the other is an on-the-spot response to mobilize children’s enthusiasm in class*”.

One other finding is of interest. The category knowledge of the act of teaching ranked high both for novice and experienced teachers. However, a closer investigation indicates contrasting perspectives between the two groups. The novice teachers showed a negative tendency, whereas the experienced teachers were positive. This pattern aligns with previous research in EFL or ESL education [8,10,13]. With regard to the teaching objectives, the experienced teachers had the ability to let the teaching process revolve around the objectives, whereas the novice teachers frequently struggled to remember or complete the set objectives in class. For instance, Teacher S explained: “*I designed this lesson with two goals. However, the second one was forgotten and not finished in class*”. The experienced teachers may also modify the teaching process or objectives in response to children’s interactions, facial expressions, and involvement. However, when children developed fresh curiosities during the lesson, the novice teachers largely adhered to the lesson plans in order to finish them promptly: “*I hope the children go back to seats as quickly as possible. The number cards are very thin, and they may tear them. Then my course may not be able to continue. I don’t want them to keep exploring*” (Teacher S).

#### 4.2.3. Implications

The study has several implications for research and practice. The significance of GPK for ECE EFL teachers was accounted for systematically in this study. Some dynamic components, such as the act of teaching, were included in the domain of GPK. The dynamic relationships among some components of GPK, such as how to address individual developmental differences and how to use child-appropriate instructional methods, have also been partially explored and analyzed. The complexity within the intercorrelations of GPK components needs further investigation [11,43,44]. Professionals in the education field in China and other EFL settings can leverage these recognized categories and subcategories of GPK to enhance knowledge-based ECE teacher education programs and curricula. Courses relating to the categories of GPK, such as knowledge of child development and knowledge of classroom management, could be essential components of both pre-service and in-service ECE EFL teacher education programs. Additionally, these programs should offer courses on developmentally appropriate assessment practices for young English learners. The stimulated recall protocol, used by the researchers, can be considered a form of self-reflection. These reflections provide teachers with insights into the professional knowledge underlying their classroom behaviors. Prioritizing the inclusion of experienced teachers is also necessary, as they can provide vital guidance and support to help novice teachers effectively apply their knowledge.

## 5. Conclusions

In conclusion, this study used Shulman’s theory of teacher knowledge and König et al.’s conceptual framework of GPK to construct a model of GPK for ECE EFL teachers [1,4]. The investigation revealed that GPK plays a crucial role in shaping classroom behaviors among ECE EFL teachers and consists of four categories and ten subcategories. Notably, novice teachers shared similarities with experienced teachers in a number of these categories, subcategories, and dominant subcategories. However, significant differences were also observed: the subcategory how to use child-appropriate instructional methods was more relevant for experienced teachers but not for novices; novice teachers discussed knowledge of classroom management more than experienced teachers; and the novice teachers showed negative feelings, while the experienced teachers were positive, toward the category of knowledge of the act of teaching. These findings offer valuable insights for designing ECE teacher education programs, particularly those focused on early English language learning and teaching, emphasizing the need for programs that are both knowledge-based and experience-informed.

## Figures and Tables

**Figure 1 behavsci-14-00526-f001:**
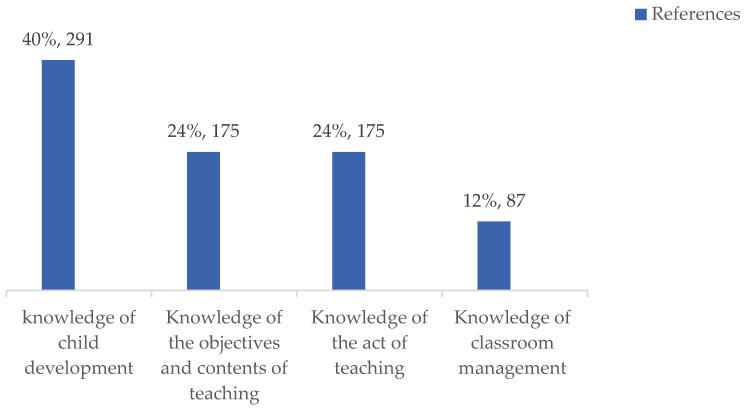
Categories of ECE EFL teachers’ GPK underpinning classroom behaviors.

**Figure 2 behavsci-14-00526-f002:**
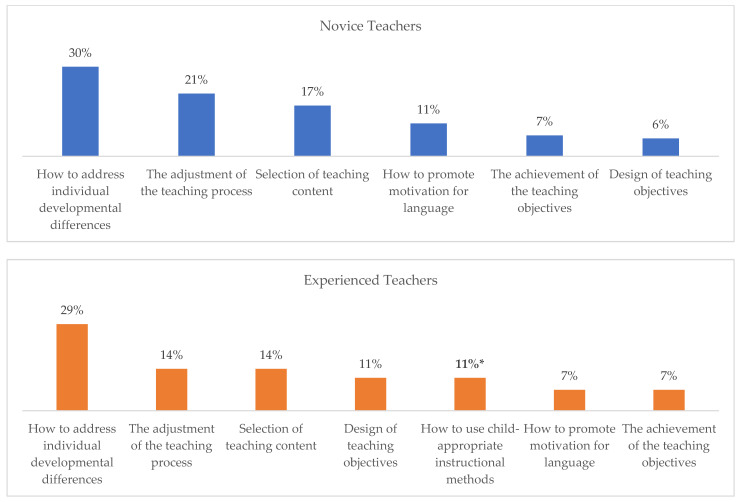
Comparisons of dominant GPK subcategories between novice and experienced ECE EFL Teachers. Note. * Prevalent in experienced teachers but not in novice teachers.

**Table 1 behavsci-14-00526-t001:** Characteristics of the six sampled teachers.

Case	Name	Major	Years Teaching English to Young Children	English Proficiency ^1^	Degree	Kindergarten
1	Teacher H	English	2	TEM-8	Bachelor	A
2	Teacher E	English	9	TEM-8	Bachelor	B
3	Teacher S	ECE	1.5	CET-4	Bachelor	C
4	Teacher R	ECE	10	CET-4	Diploma	C
5	Teacher Z	ECE	25	None	Diploma	D
6	Teacher W	ECE	16	None	Diploma	D

^1^ CET-4, the College English Test Band 4, equivalent to CEFR B1, is a recognized EFL test for non-English majors in China; TEM-8, the Test for English Majors Grade 8, equivalent to CEFR C1+, is a recognized EFL test for English majors in China. Teacher Z and W began their related work after receiving systematic training in English and English teaching for young children following their employment.

**Table 2 behavsci-14-00526-t002:** Interview schedule.

Areas to Explore	Questions
Teaching materials	✧ From my observation, I have seen you used textbooks or picture books to teach. What makes you choose these materials? ✧ Why do you use them, what aims do you want to achieve? ✧ Do you often use them? Why?
Teaching aids	✧ From my observation, I have seen you used flash cards or realia to teach. What makes you choose these materials? ✧ Why do you use them, what aims do you want to achieve? ✧ Do you often use them? Why?
Teaching activities	✧ From my observation, I have seen you used games, song, worksheets and so on to teach. What makes you choose these activities? ✧ Why do you use them, what aims do you want to achieve? ✧ Do you often use them? Why?
Evaluating and rewarding techniques	✧ From my observation, I have seen you used stickers, cards and so on to reward the children. Why do you use them? Do you often use them? Why? ✧ I also observe that you use different ways to assess children’s achievement in the class, for example, observing children’s performance in the class and taking notes of them. Why? Do you often use them? Why?
Integration of curriculum	✧ Does this English lesson have relationships with other content lessons at kindergartens? If yes, what are the relationships? ✧ Why do you construct the relationships? ✧ Do you often integrate the contents of other curriculum into your English lessons? Why?
English usage	✧ From my observation, you use all English to teach in the class (or not). Why? ✧ You also let children speak all English in classes (or not). Why?

Note. During the stimulated recall, the questions were asked in Chinese, and the participants’ answers were also in Chinese.

**Table 3 behavsci-14-00526-t003:** Coding system of the study.

Category (References, Percentage/Total: 728)	Subcategory (References, Percentage/Total: 728)	Examples
Knowledge of child development (291, 40%)	✧ How to use child-appropriate activities (22, 3%) ♦ teacher-whole class ♦ corner activities ♦ teacher-individual child	*After class, I will focus on guiding this child. However, in class, I could not always give my attention to this child. The other children were still waiting.*
	✧ How to address individual developmental differences (211, 29%) ♦ language development ♦ English learning experience ♦ cognitive and motor development ♦ emotional and social development ♦ age characteristics	*Some children could directly say ‘puppy,’ some needed to be reminded of the word ‘puppy’, and some had no response even after my reminding them again and again.*
	✧ How to use child-appropriate instructional methods (58, 8%) ♦ Nine child-appropriate instructional methods	*Some children learn faster and raise their hands positively. I do not name them. I say ‘Ok, close your eyes, close your mouth’. I name those who are not raising their hands or who are not positive.*
Knowledge of the objectives and contents of teaching (175, 24%)	✧ Selection of teaching content (109, 15%) ♦ Star English ♦ Kindergarten Happy English	*We developed a set of teaching materials for K1, K2 and K3. My kindergarten is affiliated with a foreign language university and has such resources.*
	✧ Design of teaching objectives (66, 9%) ♦ the relationships among the different teaching objectives ♦ principles for designing the objectives	*Children can perceive and sing the song Autumn, which is a cognitive objective; children can say the words with accurate pronunciation, which is an ability objective; children can feel the joy of the games, which is an emotional objective.*
Knowledge of the act of teaching (175, 24%)	✧ The achievement of the teaching objectives (51, 7%) ♦ whether or not the teaching process revolved around the objectives	*I designed this lesson with two goals: to master the word ‘dog’ and to master the sentence ‘This is a dog’. The first one was finished in class. However, the second one was forgotten and not finished in class.*
	♦ The adjustment of the teaching process (124, 17%) ♦ adjust the teaching process according to children’s reactions in class	*When reviewing the words [duck, sheep, chick, cat, dog] and the sentence [I like…] taught before, I found no responses from the children. I adjusted the goals lower, as long as they could say the word ‘banana’.*
Knowledge of classroom management (87, 12%)	✧ How to manage student behavior (22, 3%) ♦ establishing rules and routines ♦ making use of instructions ♦ using the chant	*If I want the children to sit down and listen carefully, I will use instructions, such as ‘One, two, three, four, four, four!’.*
	✧ How to gain and maintain student attention (7, 1%) ♦ stickers, games, eye contact, integration of scientific elements and design of instructional activity	*This is a K3 class. I can put in some scientific elements to stimulate their curiosity.*
	✧ How to promote motivation for language (58, 8%) ♦ finding children’s interests, games, stickers and cards, peers, verbal or gesture encouragement, competition games, English Awards Ceremonies, and songs	*Except for “Give me five”, there are “Give me hug”, “Give me kiss”, “Give me one finger”. All of these are OK.*

**Table 4 behavsci-14-00526-t004:** Comparisons of GPK categories and subcategories between novice (N = 2) and experienced teachers (N = 4), and frequency (in %) of each category and subcategory.

GPK Categories and Subcategories	Novice Teachers				Experienced Teachers					
	H	S	ALL		E	R	Z	W	ALL	
**Knowledge of child development**				33						44
1. How to use child-appropriate activities	1	2	1		5	5	0	0	4	
2. How to address individual developmental differences	36	18	30 ^1^		39	31	16	18	29 ^1^	
3. How to use child-appropriate instructional methods	1	4	2		6	16	13	5	11 ^3^	
**Knowledge of the objectives and contents of teaching**				23						25
4. Selection of teaching content	16	18	17 ^3^		10	19	11	8	14 ^2^	
5. Design of teaching objectives	6	5	6 ^6^		14	9	5	23	11 ^3^	
**Knowledge of the act of teaching**				28						21
6. The achievement of the teaching objectives	7	9	7 ^5^		4	4	20	8	7 ^4^	
7. The adjustment of the teaching process	17	30	21 ^2^		12	10	28	18	14 ^2^	
**Knowledge of classroom management**				17						10
8. How to manage student behavior	4	5	5		4	0	5	8	3	
9. How to gain and maintain student attention	2	0	1		0	1	0	0	0	
10. How to promote motivation for language	11	11	11 ^4^		7	6	3	15	7 ^4^	
Total number of references	118	57	175		104	160	64	40	368	

Note. ALL = Data collapsed across each group’s all teachers. Superscripts specify the rank of each group’s most frequently reported subcategories.

## Data Availability

The data are available upon request from the first author.

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
