# Peer review of "The General Pedagogical Knowledge Underpinning Early Childhood Education Teachers’ Classroom Behaviors Who Teach English as a Foreign Language in Chinese Kindergartens"

_behavsci, 2024, doi:10.3390/bs14070526_

Round 1

Reviewer 1 Report

Comments and Suggestions for Authors

This is an interesting study on a topic of growing interest. It reveals some insights into the teaching of EFL in Chinese ECE settings. I would have been interested to read about comparisons between the English teachers and ECE teachers but the sample was small for any conclusions to be drawn. This is of interest as there is a tendency to hire English speakers who are not ECE trained. Perhaps that can be a next study.

The conclusions could be improved - they read like a list of results rather than conclusions. 

Comments on the Quality of English Language

There are a few typos that need correcting but overall English is fine. 

Author Response

Comments to the Author

--This is an interesting study on a topic of growing interest. It reveals some insights into the teaching of EFL in Chinese ECE settings. I would have been interested to read about comparisons between the English teachers and ECE teachers but the sample was small for any conclusions to be drawn. This is of interest as there is a tendency to hire English speakers who are not ECE trained. Perhaps that can be a next study.

Thank you for your comment. The two authors have together conducted a few studies on ECE EFL teachers’ CK, GPK and PCK. We found that CK is significantly influenced by their majors(English or ECE), as shown in another study. GPK is significantly affected by their experiences. Thus, teaching experience is the focus of the current paper.

--The conclusions could be improved-they read like a list of results rather than conclusions.

Thank you for your comment. In the revised version, we rephrased the conclusions to make them more concise and precise, rather than merely listings the results or findings. Please check 5. Conclusions, p.16.

Reviewer 2 Report

Comments and Suggestions for Authors

I have read this article carefully since it is a very interesting work. Authors have made a good research work. It is hard to find teachers that allow other teachers to review their practice, and this is an added value to this research. The instrument is well design and implemented. Data collection process is correct. Literature review is updated. Authors must check citation within the text as well as the references as it does not meet the requirement of the Journal.

Additional comments:

The main question addressed in this paper is the qeneral pedagogical knowledge that English as a foreign language teacher in China have and the similarities and differences in GPK categories and subcategories between the experienced and the novice teachers in China.

I do consider this topic relevant to improve initial teachers training, to design teachers lifelong training or to design teachers performance indicators for evaluation instruments.

It is a live observation which is relevant for the community. Not always we have the chance to observe live classroom practice. It is true that the fact of having observer in the class can affect in some way the normal functioning of the class. If affecting variables are controlled this adds value to the teaching practice.

My concern is on how having an outsider in the class can compromise or have an impact on the results as children and even teachers might not behave as they do without observers.

Conclusions are consistent.

References should be corrected.

Author Response

Comments to the Author

--I have read this article carefully since it is a very interesting work. Authors have made a good research work. It is hard to find teachers that allow other teachers to review their practice, and this is an added value to this research. The instrument is well design and implemented. Data collection process is correct. Literature review is updated. Authors must check citation within the text as well as the references as it does not meet the requirement of the Journal.

Thank you for your comment. In the revised version, we thoroughly checked the references, and corrected some errors, such as placing the reference numbers before punctuation and deleting the unnecessary page numbers, please check the entire paper.

Additional comments:

--The main question addressed in this paper is the general pedagogical knowledge that English as a foreign language teacher in China have and the similarities and differences in GPK categories and subcategories between the experienced and the novice teachers in China.

I do consider this topic relevant to improve initial teachers training, to design teachers lifelong training or to design teachers performance indicators for evaluation instruments.

It is a live observation which is relevant for the community. Not always we have the chance to observe live classroom practice. It is true that the fact of having observer in the class can affect in some way the normal functioning of the class. If affecting variables are controlled this adds value to the teaching practice.

My concern is on how having an outsider in the class can compromise or have an impact on the results as children and even teachers might not behave as they do without observers.

Conclusions are consistent.

Thank you for your comment. To minimize the effects on teachers and children, the first author scheduled a familiarization period with preliminary visits before formal videotaping. Additionally, the first author sat in a remote corner of the classroom to avoid affecting the teacher’s instruction and the children’s learning throughout the study. In the revised version, these measures are described in section 2.2.1, Videotaped Classroom Observation. Please check this on p.6.

Reviewer 3 Report

Comments and Suggestions for Authors

In the literature review, general pedagogical knowledge (GPK) and its components are discussed comprehensively, although some references seem dated, and it is necessary to incorporate more recent and updated literature, particularly in the context of early childhood education and the teaching of English as a Foreign Language (EFL).To strengthen the theoretical framework, relevant theories or models related to teacher knowledge and GPK in these contexts should be discussed. The methodology section is well-described for data collection, but could contain additional information about coding procedures and strategies to ensure trustworthiness. The findings present valuable GPK categories and subcategories, but the discussion section can be extended to interpret these in relation to existing literature, compare to previous GPK studies in EFL or early childhood contexts, and provide more in-depth analysis of novice-experienced teacher differences and implications. To ensure the manuscript reflects current knowledge and captures the latest developments in the fields of GPK, early childhood education, and English as a second language, the reference list should contain an increased proportion of recent sources (only 8 out of the 48 references were published or accessed after 2020). To improve the quality and impact of the manuscript, the authors must address these points so as to ensure theoretical grounding, methodology rigor, and discussion of findings are enhanced.

Comments on the Quality of English Language

Although some phrasing could be improved with a professional proofreading or editing, the English language used in the manuscript is clear, academic, and appropriate for a research paper.

Author Response

Comments to the Author

--In the literature review, general pedagogical knowledge (GPK) and its components are discussed comprehensively, although some references seem dated, and it is necessary to incorporate more recent and updated literature, particularly in the context of early childhood education and the teaching of English as a Foreign Language (EFL). To strengthen the theoretical framework, relevant theories or models related to teacher knowledge and GPK in these contexts should be discussed.

Thank you for your comment. In literature review section of the revised version, we incorporated more recent and updated literature and deleted the outdated references. We also strengthened the relevant framework or models related to GPK in EFL context and ECE contexts, please check pp.2-3. Additionally, we strengthened the review of comparative studies on novice and experienced EFL/ESL teachers’ GPK, please check pp.3-4. These revisions provide a stronger basis for the discussion section.

--The methodology section is well-described for data collection, but could contain additional information about coding procedures and strategies to ensure trustworthiness.

Thank you for your comment. In the revised version, we added the total number of references for all GPK categories, along with the numbers for each category and subcategory to make the coding system more precise. Please check Table 3. Additionally, we added some sentences in 3.1 to describe how to calculate the percentage for each category. Please check 3.1, p.10.

--The findings present valuable GPK categories and subcategories, but the discussion section can be extended to interpret these in relation to existing literature, compare to previous GPK studies in EFL or early childhood contexts, and provide more in-depth analysis of novice-experienced teacher differences and implications.

Thank you for your comment. In discussion section of the revised version, we provided a more in-depth analysis of the structure of ECE EFL teachers’ GPK and the differences between novice and experienced teachers, comparing our findings with existing literature. We highlighted similar and differing findings, and suggested directions for future research. Please check 4.1, p.14, and 4.2, p.15.

--To ensure the manuscript reflects current knowledge and captures the latest developments in the fields of GPK, early childhood education, and English as a second language, the reference list should contain an increased proportion of recent sources (only 8 out of the 48 references were published or accessed after 2020). To improve the quality and impact of the manuscript, the authors must address these points so as to ensure theoretical grounding, methodology rigor, and discussion of findings are enhanced.

Thank you for your comment. In the revised version, we incorporated more recent and updated literature and deleted the outdated references, especially in the literature review section and the discussion section. This indeed helps us a lot to capture the latest developments in the fields of GPK, EFL, and ECE. Please check References, pp.17-19.

Reviewer 4 Report

Comments and Suggestions for Authors

First of all, thank you for your work. I was pleasantly surprised by its methodological design/development, especially concerning the work carried out between the researchers (i.e., authors) themselves along with ECE teachers, as well as the discussion on the results. Therefore, I believe the study should be published, but it also needs to be reviewed, taking into account the 'recommendations for authors' pointed out (e.g., sources and recent scholarhip) above, as well as the comments integrated into the research paper itself.

Author Response

We are thankful for the comments provided by the reviewer that help us to better our manuscript. In the revision, we have taken all these comments into full consideration. The words in italics are our responses to the comments. Based on the comments from other reviewers, we have revised the manuscript. 

Round 2

Reviewer 3 Report

Comments and Suggestions for Authors

1. It would be beneficial to summarize the key findings and their significance more clearly in the conclusion section.

2. While implications for teacher education are mentioned, specific recommendations regarding how the GPK framework can be applied to ECE EFL teacher preparation programs could be provided.

3. Researchers' positionality and reflexivity could be briefly discussed, acknowledging their subjective lens and roles in the qualitative analysis process.

4. The limitations regarding lack of assessment knowledge could be discussed, as well as why this dimension did not emerge from the data for these ECE EFL teachers.

5. There are some sections that could be trimmed to make the document more concise, such as parts of the literature review that are less directly applicable.

Comments on the Quality of English Language

The writing is clear, precise, and academic in style, relatively appropriate for a scholarly journal.

Author Response

Comments to the Author

  1. It would be beneficial to summarize the key findings and their significance more clearly in the conclusion section.

Thank you for your comment. In the revised version, we summarized the key findings and clarified their significance by adding a few sentences. Please check 5. Conclusions, p.16-17(Highlighted in yellow).

  1. While implications for teacher education are mentioned, specific recommendations regarding how the GPK framework can be applied to ECE EFL teacher preparation programs could be provided.

Thank you for your comment. In the revised version, we added the term "ECE EFL" into some sentences to make the implications more specific to this area. We also included some insights regarding the dimension of knowledge of assessment for the ECE EFL field. Please check 4.2.3 Implications, p.16 (Highlighted in yellow).

  1. Researchers' positionality and reflexivity could be briefly discussed, acknowledging their subjective lens and roles in the qualitative analysis process.

Thank you for your comment. In the revised version, we added some sentences to the Implications section to highlight the significance of the stimulated recall protocol used by the researchers for teachers’ professional development. Please check 4.2.3 Implications, p.16 (Highlighted in yellow).

  1. The limitations regarding lack of assessment knowledge could be discussed, as well as why this dimension did not emerge from the data for these ECE EFL teachers.

Thank you for your comment. In the revised version, we added some sentences into this part to discuss the limitations of the qualitative method, such as the small sample size. Please check 4.1, p.14(Highlighted in yellow).

  1. There are some sections that could be trimmed to make the document more concise, such as parts of the literature review that are less directly applicable.

Thank you for your comment. In the revised version, we trimmed the Section, 1.1. "Teaching English to Young Children in China, " by deleting several sentences and references to make the literature review more concise, please check 1.1, p.4 (Highlighted in yellow).

Round 3

Reviewer 3 Report

Comments and Suggestions for Authors

Dear authors, thank you for revising your manuscript according to the comments. The literature review on GPK in EFL/ESL contexts is comprehensive, but there still could be more discussion on how the ECE context might differ or require additional considerations. The research gap and rationale for the study could be stated more explicitly and succinctly. By explicitly stating the research gap and rationale for the study, the authors can highlight the need for investigating the application of GPK in the ECE context. This would address the lack of discussion regarding potential differences and additional considerations in this specific educational setting, contributing to a more comprehensive understanding of GPK in EFL/ESL contexts. Since only a few references were published after 2020, it would be appropriate to update the references cited. In addition, a visual representation of the findings would assist readers in gaining an understanding of the findings more easily, as the current version makes it difficult for the reader to comprehend the key findings.

Comments on the Quality of English Language

Despite some tightening and polishing of language in certain sections, the manuscript makes use of clear, appropriate academic English.

Author Response

We are thankful for the comments provided by the reviewer that help us to better our manuscript. In the revision, we have taken all these comments into full consideration. The words in italics are our responses to the comments.

Reviewer-3

Comments to the Author

  1. Dear authors, thank you for revising your manuscript according to the comments. The literature review on GPK in EFL/ESL contexts is comprehensive, but there still could be more discussion on how the ECE context might differ or require additional considerations. The research gap and rationale for the study could be stated more explicitly and succinctly. By explicitly stating the research gap and rationale for the study, the authors can highlight the need for investigating the application of GPK in the ECE context. This would address the lack of discussion regarding potential differences and additional considerations in this specific educational setting, contributing to a more comprehensive understanding of GPK in EFL/ESL contexts.

Thank you for your comment. In the revised version, we added several sentences to the literature review to explicitly and succinctly state the research gap in the ECE EFL context. Please check pp.2-3 (Highlighted in yellow).

  1. Since only a few references were published after 2020, it would be appropriate to update the references cited.

Thank you for your comment. In the revised version, we updated a few references and included some published after 2020, please check p.19. Given the limited research on ECE EFL teacher knowledge, we retained many classic and highly-cited studies on EFL/ESL teacher knowledge, such as those by Tsui and Gatbonton, as well as on early English learning and teaching by Enever, Murphy, Nunan, and Butler.

  1. In addition, a visual representation of the findings would assist readers in gaining an understanding of the findings more easily, as the current version makes it difficult for the reader to comprehend the key findings.

Thank you for your comment. In the revised version, we added two figures to help readers understand the key findings more easily. Please check Figure 1 on p.10, and Figure 2 on p.14(Highlighted in yellow).